# Soil desiccation of different microtopographies on a slope in the loess area of northern Shaanxi, China

Qingping Gou[1,2,3*], Qingke Zhu[4], Juan Chen[5]

1 School of Life Sciences (School of Ecological Forestry), Mianyang Normal University, Mianyang, China, 2 Forest Ecology and Conservation in the Upper Reaches of the Yangtze River Key Laboratory of Sichuan Province, Mianyang Normal University, Mianyang, China, 3 Engineering Research Center for Forest and Grassland Disaster Prevention and Reduction, Mianyang Normal University, Mianyang, China, 4 School of Soil and Water Conservation, Beijing Forestry University, Beijing, China, 5 College of Architecture, Changsha University of Science & Technology, Changsha, China

* qingpingg@mtc.edu.cn

## Abstract

The Loess Plateau is characterized by serious soil erosion, and different microtopographies are formed on the slopes under the action of gravity erosion and water erosion. Climate change and human activities have exacerbated the formation of soil dry layers on the Loess Plateau, and there are fewer studies on the soil desiccation of different microtopographies on the slopes. In this study, the different microtopographies on a slope typical of the loess area in northern Shaanxi Province were selected as the research object, using the undisturbed slope (US) as a reference, the soil moisture, soil water vertical profile distribution characteristics and soil desiccation of the 0–10 m soil layer of the different microtopographies on a slope were analyzed. The results showed that: the difference of shallow soil moisture in different microtopographies on a slope surface was not significant, the difference of deep soil moisture was significant, and the soil moisture overuse was the largest in the gully (GU), amounting to 386.36 mm, and the smallest in the ephemeral gully (EG) (131.02 mm); the GU, the sink hole (SH), and the scarp (SC) showed a trend of decreasing and then increasing with the increase of soil depth, and the platform (PL) has little overall trend of change in soil moisture with the increase of soil depth, and the soil moisture of US and EG shows the trend of "decreasing-then increasing-then decreasing" with the increase of soil depth. The drying intensity of different microtopographies on a slope surface: GU (34%)> SC (34%)> SH (23%)> PL (18%)> EH (13%)> US (12%), which may be due to a combination of factors such as microtopography altering ecological factors (e.g., soil moisture, light, etc.) and vegetation distribution patterns. In the future vegetation restoration process, the influence of microtopography should be fully considered to realize the sustainable development of forest and grass construction.

**Data availability statement:** All relevant data are within the paper and its Supporting Information files.

**Funding:** This research was financially supported by the project of Science and Technology Department of Sichuan Province (2024NSFSC0414), the 13th Five-Year National Key Research and Development Project (2016YFC0501705) and Mianyang Normal University Research Initiation Program (QD2024A02/071). The funders had no role in the study design, data collection and analysis, decision to publish, or preparation of the manuscript.

**Competing interests:** The authors have declared that no competing interests exist.

## Introduction

Soil moisture is an important limiting ecological factor in arid and semi-arid ecosystems, especially in the process of vegetation restoration in arid and semi-arid degraded ecosystems [1,2]. Soil moisture has direct or indirect effects on runoff generation, water evaporation, and SPAC (Soil Plant Atmosphere Continuum) [3].

The Loess Plateau is characterized by severe soil and water loss, and microtopographies such as PL, SC, SC, EG and GU are formed on the slopes under the effects of gravity erosion and water erosion, etc. For a detailed description of microtopography, see Ref. [4], and there have been many studies related to microtopography. For example, Ma [5] investigated the temporal variation of soil moisture in different microtopographies and the response to precipitation in the loess area by using a time series analysis method, and the results showed that soil moisture stored in EG, SC, and EG after rainfall affects the subsequent soil moisture for 4 weeks, whereas soil moisture stored in SC and US affects the soil moisture for 2 weeks. Shen [6] analyzed soil moisture trends from 0–160 cm in different microtopographies on a slope in loess areas and showed that SH had the fastest rate of change, while GU had the greatest amount of soil moisture. Yu [7] investigated the relationship between soil moisture and microtopography, and the results showed that microtopography altered the distribution pattern of soil moisture. Thompson [8] analyzed the response of microtopography to rainfall using an idealized model and showed that microtopography can increase rainfall infiltration by more than 20% compared to lacking microtopography. These studies above suggest that microtopography alters the rainfall bearing surface as well as runoff transport pathways, thereby affecting the spatial heterogeneity of soil moisture and other habitat conditions. However, these studies have focused only on shallow soil moisture in microtopography, and fewer studies have been conducted on deep soil moisture in microtopography.

The impacts of climate change and human activities have led to changes in soil hydrological processes, and the phenomenon of soil dried layer is the most obvious evidence [9–11]. The occurrence of dry soil layer is mainly due to the deep soil moisture through evapotranspiration (soil water output) and insufficient rainfall recharge (low soil water input) [12,13]. Dry soil layer phenomenon has been found in Russia [14], the Amazon [15,16], Australia [17], the southwestern United States [18], and the Loess Plateau of China [19]. The presence of dry soil layer can block plants from utilizing groundwater, especially in drought or extreme drought years, which is detrimental to plant growth and threatens regional ecological security [20]. The Loess Plateau has low rainfall, high potential evapotranspiration, and concentrated rainfall, and precipitation usually only recharges the 1 m deep soil layer [21], which results in insufficient soil moisture recharge and increasingly severe soil desiccation [20]. According to previous research, soil desiccation phenomenon exists in most areas of the Loess Plateau of China, and many scholars have conducted a lot of researches on soil desiccation phenomenon in this region in order to better recognize and understand soil desiccation, such as: different vegetation types, different land use types, and different topographic elements [22], and so on. Liang [23] showed that the

degree of soil desiccation of arbors was then greater than that of shrubs and herbs; Li [24] showed that the soil desiccation effect was smaller in agricultural land and greater in woodland. Although these studies have made important progress, fewer studies have been reported on soil desiccation effects on different microtopographies.

Therefore, this study was based on the typical different microtopographies in the loess area of northern Shaanxi as the research object, and by investigating the soil moisture in the 0–10 m soil layer of different microtopographies, the following three questions were mainly answered: (1) The soil moisture content in different microtopographies; (2) Characteristics of the vertical profile distribution of the soil moisture in different microtopographies (3) and Degree of soil desiccation in different microtopographies.

## Materials and methods

### 1. Study area

The study area is located in the northern part of the Loess Plateau, in Wuji County (36°33′33″-37°24′27″N,107°38′57″-108°32′49″E), Yan'an City, Shaanxi Province. The geomorphology of the county is loess hills and gullies, with a warm-temperate continental monsoon climate, an altitude of 1233–1809 m, an average annual temperature of 7.8°C, an average annual precipitation of 478.3 mm, an uneven seasonal distribution of precipitation, large inter-annual variations, and the rainy season concentrating in July-September, with an average annual land surface evaporation of 400–450 mm. The soil is a zonal black clayey soil stripped and extensively developed on loess parent material of loessial soil, accounting for 97.6% of the total area of the region, with light loamy texture, 20.9% water holding capacity in the field, and 4.7% wilting moisture [25,26], and the average soil bulk weight of the soil layer above 1 m was 1.3 g/cm$^3$. The Hegou watershed has been implemented to return farmland to forest since 1998, and the main vegetation type is herbaceous, with trees sporadically distributed in the gully. The local government has planted 2-year-old seedlings of Chinese pine (*Pins tabuliformis*) on the slopes since 2016 in order to quickly realize the purpose of vegetation restoration; however, the deep soil moisture, which is the concern of our study, has little effect and can be ignored.

### 2. Research methods

**2.1. Sample plot selection.** We collected soil samples in the field, which have been approved by the local Wuji County Forestry Department. On the Loess Plateau, soil erosion is serious, and under the action of water erosion and gravity erosion, etc., various kinds of broken topography are formed on the slope surface, Zhu [27] proposed the concept of microtopography on the basis of summarizing the research of the previous research, and categorized these broken topographies, and finally classified the microtopography into five kinds of microtopographies, namely, platform (PL), scarp (SC), ephemeral gully (EG), sink hole (SH), and gully (GU), and the details about the microtopographies can be found in the literature [4,28]. In order to study the soil desiccation effect of different microtopographies, we selected a typical slope in the Hegou watershed of Wuji County, which is well developed with various microtopographies, and has a good reference value for the study of soil desiccation effect under various microtopographies.

In this study, five types of microtopography, namely, platform, scarp, ephemeral gully, sink hole and gully on a typical slope were selected as the object of study, and the undisturbed slope was used as the control for the study. Firstly, the basic information of each microtopography was investigated, and the specific results are shown in S1 Table.

**2.2. soil moisture investigation methods.** Based on the survey of basic information of different microtopographies, soil moisture of different microtopographies and undisturbed slope was investigated by manual soil auger method in October 2017, May and October 2018 and 2019, and a total of five surveys were conducted, each time before and after the rainy season of each year.

For each survey, three sample points were randomly selected for sampling under each type of topography. Under each sample point, soil samples were taken at depths of 0–1000 cm, at 10 cm intervals from 0–100 cm, and at 20 cm intervals

from 100–1000 cm, for a total of 55 layers of soil samples per sample point, with three replicates per layer, for a total of 165 soil samples per sample point. At the time of taking each sample, the weight of the empty aluminum box was weighed in advance and recorded as $m_0$, when it was filled with soil, it was weighed with an electronic balance and the weight at this time was recorded as $m_1$, after taking the sample, it was brought back to the laboratory and put into the drying oven at constant temperature (100°C, 24h) until it was constant weight [29], it was taken out, and it was weighed with an electronic balance, at this time the weight was recorded as $m_2$.

**2.3. Calculation of soil moisture related indicators.** Soil moisture (SM) was calculated as:

$$SM = \frac{m_1 - m_2}{m_2 - m_0}$$

Soil water storage (SWS) is the amount of water stored in a certain thickness of soil layer, calculated by the formula [30].

$$SMS = \frac{10 \times SM \times BD \times H}{\rho}$$

In the formula, SMS for the soil water storage capacity (mm); bulk density (BD) for the soil bulk weight (g/cm³), SM for the soil mass water content (%), H indicates the thickness of the soil layer (cm), $\rho$ indicates the water density, which is taken as 1.0 g/cm³.

In order to study the variation of SWS in different microtopographies and undisturbed slope, we collected undisturbed soil samples with a ring knife on different microtopograpies and undisturbed slope, and calculated the soil bulk density of different microtopographies and undisturbed slope. Details are shown in S2 Table. Due to the difficulty of sampling deeper in undisturbed soil samples [31], when we have soil water storage capacity of soil layer below 100 cm, the capacity weight is taken as the value of soil bulk density of soil layer 90–100 cm.

Available soil water storage (ASMS) represents the difference between the actual soil water storage and the soil water storage at wilting moisture and is calculated as:

$$ASMS = SMS - SMS_{WM}$$

In the formula, ASMS is the effective soil water storage (mm); $SMS_{WM}$ indicates the soil water storage (mm) when the soil moisture is wilting moisture.

Soil moisture overuse (SMO) represents the subtraction of the amount of water stored in the soil at the stable moisture level from the actual amount of soil water stored and is calculated by the formula:

$$SMO = SMS_{SSM} - SMS$$

Where SMO is the soil moisture overuse (mm); $SMS_{SSM}$ indicates the amount of soil water stored in soil moisture at a stable moisture (mm).

**2.4. Indicators for evaluating soil desiccation.** The lower limit water content index of the dried soil layer in the Loess Plateau is lower than the wilting moisture, while the upper limit index is lower than the soil stable moisture or the capillary fracture moisture. The stable soil moisture refers to a stable moisture value that the soil can maintain over time, which is generally 50% to 80% of the field water holding capacity, and which reflects the intermediate state of a given soil's water-holding performance [10,13].

In this study, according to the soil characteristics, which belongs to the loessial soil, the field stable moisture content was taken as 60% of the field water holding capacity (20.9%) [4,25], and the value of wilting moisture was taken as 4.7% [25].

In order to quantitatively describe the soil desiccation status of different microtopographies, the soil desiccation index (SDI) was used to analyze and compare the soil desiccation effect of different microtopographies [24,32]. The formula of SDI is as follows:

$$\text{SDI} = \left(1 - \frac{SM - WM}{SSM - WM}\right) \times 100\% = \frac{SSM - SM}{SSM - WM} \times 100\%$$

Where SDI is the soil desiccation index; SSM is the soil stable moisture, which takes the value of 12.5%; and WM is the soil wilting moisture, which takes the value of 4.7%.

According to the size of the SDI, the degree of soil desiccation is divided into six grades, and the specific classification criteria are shown in S3 Table.

**2.5. Methods of statistical analysis.** The data were analyzed and processed using software such as Microsoft Office 2016 (2015, Microsoft Corporation, USA) and IBM SPSS Statistics (2009, V. 18.0; IBM Corporation, Armonk, NY, USA). One-way ANOVA, and LSD multiple comparisons were used to analyze soil moisture differences between microtopographies, and graphing was done in Microsoft Office 2016.

## Results

### 3.1. Soil moisture of different microtopographies

The ANOVA results of soil moisture in the shallow layers (0–200 cm) and deep layers (200–1000 cm) of different microtopographies are shown in S4 Table. As can be seen from S4 Table, the difference of soil moisture in the shallow layer of different microtopographies was not significant ($P>0.05$), in which the soil moisture of the US was the largest, while the soil moisture of the EG was the smallest; which the minimum and maximum values of soil moisture both appeared in the GU, with values of 6.71% and 17.37% respectively, which indicated that the shallow soil moisture in the GU had the greatest variability; in the shallow layer, the average values of soil moisture in different microtopographies ranged from 11.39% to 12.83%, in the following order from the highest to the lowest: US>SH>SC>PL>GU>EG. In the deep layer, the differences between different microtopographies were significant ($P<0.05$), EG (11.56%) was significantly larger than SH (10.34%), SC (10.23) and GU (9.31%), and SH was significantly larger than GU ($P<0.05$); in the deep layer, the maximum value of soil moisture appeared in the US, which amounted to 15.85%, and the minimum value appeared in the GU, which amounted to 6.53%; the deep layer of soil moisture in the microtopographies was in the order of: EG>US>PL>SH>GU. Through the analysis, we found that microtopography with high soil moisture content in the shallow layer has small soil moisture content in the deep layer.

The average values of soil moisture, soil water storage, effective soil water storage and soil water overuse in the 0–10 m soil layer of different microtopographies are shown in S5 Table. The average value of soil moisture in the whole soil layer of 0–10 m was the highest in the US, which reached 11.65%, followed by the PL and EG, and the SH and SC were closer to each other, which were 10.75% and 10.65% respectively, and the lowest was in the GU; the average value of soil water storage in the different microtopographic soil layers ranged from 1241.57 mm to 1471.41 mm, in which the US was the highest, and the GU was the smallest. the effective soil water storage capacity ranged from 631.42 mm to 869.91 mm, in addition to the US, the PL and the EG was higher than 800 mm, the SC, the SH and the GU was much lower than 800 mm, in which the GU was only 631.42 mm. The highest soil moisture overuse was nearly 400 mm, and the smallest soil moisture overuse was 133.45 mm on the US; different microtopographies showed different degrees of soil moisture overuse, indicating that different microtopographies also showed different soil dry layers.

### 3.2. Characterization of soil moisture vertical profile distribution in different microtopographies

Vertical profile distribution of soil moisture in different microtopographies is shown in S1 Fig. As a whole, the soil moisture of GU, SH and SC decreases and then increases with the depth of the soil layer; the soil moisture of PL does not change

much with the increase of the depth of the soil layer, and there is a tendency for the intermediate layer of the soil moisture in the US and EG to have a high value area, and then to decrease rapidly. According to the criteria for determining the soil dry layer, the soil moisture is lower than the stable soil moisture, which is the soil dry layer. It can be seen that different microtopographies have different degrees of soil dry layer development. The soil dry layer of the GU is located in the soil layer of 60–880 cm, the soil dry layer of the SH is in the soil layer of 70–780 cm and the soil layer of 980–1000 cm, and there is a multi-layer distribution phenomenon of the soil dry layer in microtopography such as SC, PL, EG, and US. The soil dry layer of SC: 0–90 cm, 140–400 cm, 480–920 cm, 960–1000 cm; the soil dry layer of PL: 30–440 cm, 480–920 cm, 960–1000 cm; the soil dry layer of EG: 20–200 cm, 380–860 cm, 940–1000 cm; the soil dry layer of US: 30–50 cm, 160–240 cm, 500–1000 cm.

### 3.3. Degree of soil desiccation in different microtopographies

In this study, the average of five soil moisture surveys was used to calculate the soil desiccation effect of different microtopographies, based on which the soil desiccation index of different microtopographies in the whole soil profile from 0–1000 cm was calculated, and the results are shown in S6 Table. As can be seen in S6 Table, the average soil desiccation indices of the different microtopographies ranged from 12% to 34%, with the largest one in the GU (34%), and the degree of desiccation reached moderate desiccation, which indicated that the utilization of soil moisture in the GU was the largest, and all other microtopographies were mild desiccation. The thickness of the intense desiccation layer in the GU was 20 cm, and no intense desiccation layer appeared in the other microtopographies and the US.

In the severe desiccation layer, from largest to smallest, the order was GU > SH > US > SC > EG > PL; except for the SC and the EG, which had a larger moderate desiccation layer, there was not much difference in the moderate desiccation layer in the rest of the microtopography; it is worth noting that the mild desiccation layer in the PL was significantly larger; in terms of the whole thickness of the soil desiccation layer, the PL was the largest (940 cm), while the US was the smallest (650 cm).

## Discussion

### Soil moisture response to microtopography

The influencing factors of soil moisture have scale effects. At the regional scale, soil moisture is affected by factors such as rainfall, topography, and the nature of the soil itself [33–37]; at the watershed scale, soil moisture is affected by factors such as vegetation type, gully-slope dichotomy, and other factors [29]; and at the slope scale, soil moisture is affected by factors such as slope gradient, slope orientation, slope length, microtopography, and vegetation [5,7,35]. In this study, we focused on the soil desiccation of different microtopographies at the slope scale. There was no significant difference in soil moisture in the shallow layer (0–200 cm) of different microtopographies, which was slightly different from the study of Puyang et al. (2020), probably because Puyang's study used data from a single survey [38], whereas we used the average of data from multiple surveys; In addition, the Loess Plateau has deep soil layers, precipitation is the only source of soil moisture recharge, and the depth of precipitation recharge is generally is 1–2 m [21], and shallow soil moisture is susceptible to rainfall, resulting in some fluctuations.

The soil layer of the SH is loose; the cross-section of the GU is "V"-shaped, and the length, width and depth of the down-cutting scale are large; the PL can reduce the runoff flow rate; all of these microtopographic features mentioned above are conducive to the infiltration of soil moisture during rainfall and recharge more soil moisture, which is also confirmed by the results of the previous studies [6,38,39]. However, our study showed that the soil moisture in the deep layer (200–1000 cm) of different microtopographies, however, exhibited: EG > US > PL > SH > SC > GU; Microtopographies with better soil moisture conditions in the shallow layers of the SH, GU and PL, on the contrary, had lower soil moisture in the deep layer, which may be due to the fact that the better soil moisture conditions in the shallow layers of the SH, GU and

PL are favorable to the vegetation restoration process, especially the early plant settlement and growth during natural restoration, due to the influence of warm drying of the Loess Plateau [40], the vegetation absorbs the deep soil moisture through the plant root system in order to maintain its own growth [41] for the maintenance of its own life activities required, which leads to the decrease of deep soil moisture in different microtopographies. Previous studies have also shown that the vegetation cover, average height and biomass were also significantly higher in GU, SH and PL with better soil moisture conditions when fencing and sealing were adopted [42].

## Soil desiccation in different microtopographies

According to Yang [43], the emergence of soil desiccation is mainly influenced by factors such as climatic, vegetation construction, soil and topography. In recent years, against the background of rising temperatures, decreasing precipitation [40], and increasing drought in the Loess Plateau, vegetation transpiration water depletion and bare-land soil moisture evaporation are increasing, different microtopographies show different degrees of desiccation phenomena, and an moderate desiccation layer occurs in the GU. Li [24] showed that soil desiccation effects existed to different degrees in the 0–10 m soil layer in woodland, grassland and farmland; He [44] studied the soil desiccation problem in the Wangdonggou sub-watershed of the loess gully area, and found that there was a general soil desiccation layer in the watershed, and the degree of soil desiccation was as follows: woodland > orchard > grassland > farmland; Yu [45] showed that a significant negative correlation between soil desiccation index and locust (*Robinia pseudoacacia*) tree age, tree height, diameter at breast height (DBH), and biomass; these studies showed that the main influencing factor of soil desiccation phenomenon is vegetation type.

Topography mainly affects shallow soil moisture, while vegetation mainly affects deep soil moisture [46]. The development of different microtopographies on slopes changes the rain-bearing surface, leading to differences in soil moisture in different microtopographies [5], which in turn changes the distribution pattern of vegetation on different microtopographies [46], and different vegetation patterns lead to different microtopographic soil desiccation phenomena. The degree of soil desiccation in the GU is moderate, while the degree of soil desiccation in the other microtopographies is mild. This may be due to the fact that during the natural recovery of vegetation, the shallow soil moisture conditions in the GU are the best and the natural vegetation recovery is the fastest, which is caused by the fact that due to the influence of the dry climate, the vegetation utilizes more soil moisture in order to maintain its life activities. Furthermore, soil moisture is also influenced by climate change [47,48]. By the end of the 21st century, global climate risks are projected to increase significantly with rising greenhouse gas emissions [49], and the land surface will face severe aridification [50]. Relevant studies indicate that approximately (65 ± 26)% of evaporation originates from soil moisture rather than surface water [51]. These findings demonstrate that the formation of dry soil layers results from multiple contributing factors. Our conclusions represent inferences drawn based on experimental observations and previous literature, rather than definitive determinations.

Our observations have revealed varying degrees of soil desiccation across different microtopographies, for which we have proposed preliminary hypotheses. It is important to note, however, that the relationships among microtopography, vegetation parameters (biomass, canopy cover), and SDI require further validation through extensive data collection and analysis.

## Insights into the sustainability of vegetation construction

The phenomenon of soil desiccation in different microtopographies of slopes indicates that we need to introduce sufficient attention in the process of vegetation construction in the future. Wang [52] analyzed the structural features and water consumption characteristics of natural and artificial forests in the Ziwuling Mountains, and found that natural forests have formed a typical "tree-shrub-grass" complex and stable spatial layer structure, with a strong self-regulation ability, and a mild water deficit in the shallow layer, which can be recovered in time after the rainy season, and has no effect on

the development and succession of natural vegetation. This has no effect on the development and succession of natural vegetation, but the artificial forest land has serious soil desiccation phenomenon. In the process of vegetation construction in the future, we should avoid the traditional model of afforestation, which is based on the contour line and in accordance with a certain spacing between plants and rows. We should combine the theory of restoration ecology, investigate the local historical vegetation background, make full use of native plants, and match the stable structure of tree-shrub-grass complex, at the same time, and make full use of precipitation resources in order to avoid the intensification of soil desiccation and the formation of permanent soil dry layer.

## Conclusion

(1)  Based on the analysis of the mean values of the five surveys, the differences in shallow soil moisture among different microtopographies on the slope were not significant; the differences among deep soil moisture were significant. In the 0–10 m soil layer, throughout the soil profile, the phenomenon of overuse of soil moisture occurred in different microtopographies, in which the GU was the largest, followed by SH and SC, while the EG and the US were smaller.

(2) From the vertical profile distribution of soil moisture in different microtopographies, the soil moisture of GU, SH and SC showed a trend of decreasing and then increasing with the increase of soil depth, the soil moisture of PL did not change significantly with the increase of soil depth, and the soil moisture of US and EG showed a trend of "decreasing-increasing-decreasing" with the increase of soil depth.

(3) The soil desiccation index (SDI) of different microtopographies on a slope ranged from 12% to 34%, with the GU already belonging to moderate desiccation, and the SH, SC, PL, EG and US belonging to mild desiccation. The intensity of soil desiccation is GU > SC > SH > PL > EG > US, which may be due to the combination of microtopography changing ecological factors (e.g., soil moisture, light, etc.) as well as the vegetation distribution pattern. In the future vegetation construction, the influence of slope microtopography should be fully considered to realize the sustainable development of forest and grass construction.

## Supporting information

**S1 Data.**
(RAR)

**S1 Table.  Basic features of microtopography.**
(TIF)

**S2 Table.  Soil bulk density of different microtopographies.**
(TIF)

**S3 Table.  Criteria for classifying the degree of soil desiccation.**
(TIF)

**S4 Table.  Comparison of soil moisture in different microtoporaphies (Note: Different letters in the same column indicate significant differences (P<0.05)).**
(TIF)

**S5 Table.  Comparison of soil moisture storage capacity in 0–10 m soil layer of different microtopographies.**
(TIF)

**S6 Table. Soil desiccation of different micritopographies (Note: SC, SH, US, PL、GU and EG indicates scarp, sink hole, undisturbed slope, platform, gully and ephemeral gully, respectively).s.**
(TIF)

**S1 Fig. Vertical profile distribution characteristics of soil moisture in different microtopographies (SC、SH、US、PL、GU、EG、WM and SSM indicates scarpsink hole、undisturbed slope、platform、gully、ephemeral gully、wilting moisture and soil stable moisture, respectively).**
(TIF)

## Acknowledgments

The author thanks Xianglei Tian, Pengxiang Wang, Yuyan Liu, and Minghuang Shen for their contributions to the field work.

## Author contributions

**Funding acquisition:** QINGKE ZHU.

**Writing – original draft:** QINGPING GOU.

**Writing – review & editing:** QINGPING GOU, QINGKE ZHU, JUAN CHEN.

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
