## [Decision Letter · Decision Letter 0]

9 Jun 2025

Dear Dr. GOU,

Thank you for submitting your manuscript to PLOS ONE. After careful consideration, we feel that it has merit but does not fully meet PLOS ONE’s publication criteria as it currently stands. Therefore, we invite you to submit a revised version of the manuscript that addresses the points raised during the review process.

As pointed by the reviewers, the topic of this manuscript is relevant and important, but some major concerns are raised such as lack of definition of Loess Plateau, lack of justification of sampling methods, missing of some data validation, inadequate interpretation and discussion on the results, and overstatement in the conclusion. I would therefore recommend a major and careful revision based on the reviewers comments, and a further review might be needed as requested by the editors and reviewers. I hope you will find the comments helpful to improve the quality of your manuscript.

We look forward to receiving your revised manuscript.

Kind regards,

Ziming Yang, PhD

Academic Editor

PLOS ONE

Journal Requirements:

1,Please ensure that your manuscript meets PLOS ONE's style requirements, including those for file naming. The PLOS ONE style templates can be found at

4. In this instance it seems there may be acceptable restrictions in place that prevent the public sharing of your minimal data. However, in line with our goal of ensuring long-term data availability to all interested researchers, PLOS’ Data Policy states that authors cannot be the sole named individuals responsible for ensuring data access (http://journals.plos.org/plosone/s/data-availability#loc-acceptable-data-sharing-methods).

7. We note that Figure 1 in your submission contain copyrighted images. All PLOS content is published under the Creative Commons Attribution License (CC BY 4.0), which means that the manuscript, images, and Supporting Information files will be freely available online, and any third party is permitted to access, download, copy, distribute, and use these materials in any way, even commercially, with proper attribution. For more information, see our copyright guidelines: http://journals.plos.org/plosone/s/licenses-and-copyright.

Reviewers' comments:

Reviewer's Responses to Questions

**Comments to the Author**

1. Is the manuscript technically sound, and do the data support the conclusions?

Reviewer #1: Partly

Reviewer #2: Yes

Reviewer #3: Partly

2. Has the statistical analysis been performed appropriately and rigorously?

Reviewer #1: Yes

Reviewer #2: Yes

Reviewer #3: Yes

3. Have the authors made all data underlying the findings in their manuscript fully available?

Reviewer #1: No

Reviewer #2: Yes

Reviewer #3: Yes

4. Is the manuscript presented in an intelligible fashion and written in standard English?

Reviewer #1: Yes

Reviewer #2: Yes

Reviewer #3: Yes

Reviewer #1: As part of the peer review process, I would like to offer several critical observations aimed at enhancing the academic and methodological rigor of this manuscript. The topic is relevant and contributes meaningfully to the understanding of hydrology and soil conservation in the Loess Plateau region. However, there are several areas that require clarification or further examination, particularly regarding the consistency of arguments, methodological validity, representativeness of the data, and the strength of evidence supporting the conclusions. The following comments are intended as constructive feedback to help improve the manuscript prior to publication.

The text mentions that vegetation restoration has been carried out since 1998, but then states that vegetation “does not significantly affect deep soil moisture”. This contradicts the subsequent discussion that vegetation is the main cause of desiccation. In addition, there is no mention of vegetation type, age or density in each micro-topography, which is one of the key variables.

I felt the author needed to display a distribution map of the sampling sites or a detailed topographic map showing the spatial position of the micro-topography.

The authors only used one slope site with multiple micro-topography types in one watershed (Hegou). It is not adequately explained whether this site is representative for the entire Loess Plateau. This risks generalization bias. It is also not statistically explained whether the variation between observation points (n=3 per micro-topography) is sufficient to ensure data reliability and validity. The sample size is very limited given the high spatial variability of soil moisture. How do the authors respond to this?

The authors did not confirm inter-annual consistency or significant seasonal influences, even though the survey was conducted before and after the rainy season in 3 different years.

For depths >100 cm, bulk density is equated to the 90-100 cm layer. This assumption tends to be methodologically weak, as bulk density can increase at depth due to soil pressure. This may cause estimation errors in the calculation of soil water storage (SWS).

The authors adopted soil desiccation index (SDI) with universal thresholds (wilting point 4.7% and SSM = 60% of field capacity). However, there is no local validation that this parameter is suitable for all micro-topographic soil types in the study site. This has the potential to create over/underestimation in the classification of drought degree.

The conclusions made by the authors tend to Exceed the Evidence. For example, the claim that soil desiccation is “caused by vegetation distribution patterns” has not been supported by statistical analysis of the relationship between vegetation and the SDI index. No regression or correlation is shown. In addition, the description of “natural vegetation recovery” in GU as a cause of soil desiccation is speculative, with no quantitative data on actual root biomass or vegetation structure. How do the authors respond to this?

There is no mention of the influence of previous land management, land use history, or potential disturbances (such as human activities, grazing, etc.) that could affect soil moisture or vegetation. I recommend that this is disclosed, albeit briefly.

The overuse soil moisture value in GU is very high (386.36 mm), but there is no evaluation of whether this value is hydrologically realistic under conditions of 478.3 mm annual rainfall. In addition, the 940 cm thickness of the desiccation layer in PL is quite astonishing, but there is no detailed discussion on how the layer formed to such a depth, even though PL is described as an area with high infiltration and low erosion.

I did not find a section that explicitly discusses the limitations of the study in the conclusion, even though many large assumptions were used.

Reviewer #2: The authors take on the question of how various topographical features contrbiye to the problem of soil erosion in a loess environment at Shaanxi Province of China. I agree that the abstract shpuld reflect the core of the research however, in its current form the abstract contains too many abbreviations which make it difficult to follow. May I suggest at least removing the abbreviations from the abstract. In the abstract, authors use “decreasing-then increasing-then decreasing ” to define a patterned change in soil moisture. It may be more practical to use the term 'see-saw'.

I recommned that authors geographically define the Loess Plateau the first time that they refer to it in the Introduction. I do not see a map of the region or China as a reference to the research area. These should be added to the manuscript. Please revise the sentence on Lines 46-47.I would replace the word 'security' with 'sustainability' in Line 69 and omit the word 'typical' in Line 81. Please write the three points between Lines 83 and 86 in question form.

It will be beneficial to provide a map of sampling points across the research area. Although it is a commonly used method, I believe that the general audience will benefit from explaining what ANOVA is and what it measures. In the section 3.2 the Figure 2 forms the core of this manuscript. As much as the authprs build a convincing argument, I suspect that the general audience does not know the Loess Plateau. I recommend that authors prepare an illustration / sketch (if not an aerial imagery of their research area) that shows all the various microtopographical elements and how soil moisture levels differe by depth. A bar graphic showing the amount of moisture by depth may be superimposed on this illustration.

Reviewer #3: “Soil desiccation of different microtopographies on a slope in the loess area of northern Shaanxi, China” titled manuscript explores climate change and human activities on soil erosion. Northern Shaanxi Province in China is selected in the application. The subject is very important and the study is valuable in terms of soil erosion applications in hydrology but the novelty of the study is emphasized insufficiently. Some suggestions and comments to the authors are presented below:

1. In Abstract, the applied methodology should be explained in detail.

2. What is the novelty of the paper? Supported and related studies should be strongly presented in the paper by emphasizing the novelty of the paper. What are main differences between previous studies and this paper?

3. More performance metrics can be used to assess application results. The application results don’t show a good performance. For this aim, RMSE, R-squared, RSR, volume error, etc. can be calculated.

4. Latest studies about climate change should be discussed in the paper. See suggested papers, 10.2166/wcc.2024.207 ; 10.1016/j.catena.2021.105633 …

5. Is it only a case study or can suggested methodology be applied to world-wide similar problems in river basins with plateau, mountain and monsoon climates?

6. As one important step of the study, the statistical characteristics of used data should be presented in detail. The statistical properties as coefficient of variation, confidence intervals, boxplots for outlier data, trends, distribution characteristics and median, etc. of used data should be given in a table.

**Do you want your identity to be public for this peer review?** For information about this choice, including consent withdrawal, please see our Privacy Policy

Reviewer #1: **Yes: ** Amiruddin Akbar Fisu

Reviewer #2: **Yes: ** Bülent Arıkan

Reviewer #3: **Yes: ** Halil Ibrahim Burgan

---

## [Author Response · Author response to Decision Letter 1]

21 Jul 2025

Dear Editors and Reviewers:

Thank you very much for your attention and the reviewers’ evaluation and comments on our paper “Soil desiccation of different microtopographies on a slope in the loess area of northern Shaanxi, China” (PONE-D-25-13266R1). Those comments are all valuable and extremely helpful for revising and improving our paper, as well as providing significant guidance for our research. We have studied comments carefully and have made correction which we hope meet with approval. We invite a native English scientist of relative research to polish and edit our wording and language, the details of the changes can be found in the manuscript. Besides, our current manuscript followed the journal formatting guidelines of PLOS One. We hope the latest version of the manuscript could meet the journal’s standard. Revised portion are marked in red throughout the revised manuscript. The main corrections in the paper and the Responses to the reviewer’s comments are as flowing:

NOTE: All the Page and Line numbers where revisions were made refer to the Manuscript and Highlight with marked changes (Revised Manuscript (marked up).docx). The Revised Manuscript (clean copy) Version was the same version of the Revised Manuscript (marked up) with cleaned from all the marks.

Soil desiccation of different microtopographies on a slope in the loess area of northern Shaanxi, China PONE-D-25-13266R1

As part of the peer review process, I would like to offer several critical observations aimed at enhancing the academic and methodological rigor of this manuscript. The topic is relevant and contributes meaningfully to the understanding of hydrology and soil conservation in the Loess Plateau region. However, there are several areas that require clarification or further examination, particularly regarding the consistency of arguments, methodological validity, representativeness of the data, and the strength of evidence supporting the conclusions.

Response: Thank you very much for your comments and suggestions.

The following comments are intended as constructive feedback to help improve the manuscript prior to publication.

Response: Thank you very much for your comments and suggestions.

1. The text mentions that vegetation restoration has been carried out since 1998, but then states that vegetation “does not significantly affect deep soil moisture”. This contradicts the subsequent discussion that vegetation is the main cause of desiccation. In addition, there is no mention of vegetation type, age or density in each micro-topography, which is one of the key variables.

Response: Thanks for your comments. In order to control the serious soil erosion on the Loess Plateau, our governmental department implemented the Returning Farmland to Forestry and Grassland Forestry Ecological Project in 1998, and implemented a vegetation restoration program mainly based on artificial afforestation in sloping arable land above 25°, supplemented by natural restoration. In order to compare the differences between natural and artificial restoration, our research team selected the Hegou watershed as a sample site for natural restoration and the Chaigou watershed as a sample site for artificial restoration, and carried out a watershed pairing study, and our study showed that restoring vegetation entirely by nature takes at least 50 years later than artificial restoration. Since 2016, our research program has ended and the local governmental department planted 2-year old oil pine saplings in the Hegou watershed in the same year in order to fulfill the reforestation task of the forestry department for the purpose of rapid vegetation restoration. The sampling time of our study was in 2017, 2018 and 2019, and the focus of our study was also concerned with deep soil moisture, and the effect of planted creosote bush saplings on deep soil moisture was, in this study, negligible, which was the assumption made when our study was conducted. Our group conducted a study on the plant characteristics of different slope microtopographies, as described in the following references.

Reference:

Wang J, Zhu QK, Qin W, Zhang HZ, Yun L, Xie J, Kuang GM. Differentiation of vegetation characteristics on slope micro-topography of fenced watershed in loess area of north Shaanxi province, northwest China. Chinese Journal of Applied Ecology. 2012; 23: 694-700. (In Chinese)

2.I felt the author needed to display a distribution map of the sampling sites or a detailed topographic map showing the spatial position of the micro-topography.

Response: Thank you very much for your comments and suggestions. A map of the distribution of the different microtopographies of our study and a detailed topographic map can be seen in Fig. 1. The different microtopographies are located on the same side of the slope in the Hegou watershed.

3.The authors only used one slope site with multiple micro-topography types in one watershed (Hegou). It is not adequately explained whether this site is representative for the entire Loess Plateau. This risks generalization bias. It is also not statistically explained whether the variation between observation points (n=3 per micro-topography) is sufficient to ensure data reliability and validity. The sample size is very limited given the high spatial variability of soil moisture. How do the authors respond to this?

Response: Thank you very much for your comments and suggestions. The research scope of different microtopography is the slope scale. Our group found 39 watersheds in Wuji County (a representative area of Loess Plateau) by establishing a 10 km grid point method. Through investigation, summarization and analysis, we creatively put forward the concept of slope microtopography: the local topography with a range of 1 m2 or more, which has different shapes and sizes due to the action of soil erosion and other effects and the difference of habitat conditions such as soil moisture, soil nutrients and so on. According to the characteristics of microtopography, there are five types of microtopography: shallow gully, cut gully, collapse, slow platform and steep canyon. Based on the scale of our study, together with our soil moisture surveys in 3 years, 2017, 2018 and 2019, and the depth of sampling (10 m), we chose 3 sample sites for each sampling.

4.The authors did not confirm inter-annual consistency or significant seasonal influences, even though the survey was conducted before and aSer the rainy season in 3 different years.

Response: Thank you very much for your comments and suggestions. Our study focuses on the deep soil moisture in different slope microtopographies, and the seasonality of rainfall affects the soil moisture dynamics. According to the results of long-term positional observations by previous authors, the influence of rainfall on soil moisture at the inter-annual or seasonal scales is mainly affected by the infiltration depth of rainfall, and with reference to the existing studies (see the following literature), in the Loess Plateau region, the annual average infiltration depth of rainfall is around 1 In the Loess Plateau region, the average annual infiltration depth of rainfall is about 1 meter, which does not have a large impact on deep soil moisture.

Reference

Liu, B., Shao, M. Response of soil water dynamics to precipitation years under different vegetation types on the northern Loess Plateau, China. J. Arid Land 8, 47–59 (2016). https://doi.org/10.1007/s40333-015-0088-y.

5.For depths >100 cm, bulk density is equated to the 90-100 cm layer. This assumption tends to be methodologically weak, as bulk density can increase at depth due to soil pressure. This may cause estimation errors in the calculation of soil water storage (SWS).

Response: Thank you very much for your comments and suggestions. Measurement of soil capacity needs to take in situ soil samples, the deeper the soil depth, the more difficult it is to take in situ soil samples, based on the results of the study in the Loess Plateau and the existing reports, the soil capacity in the soil layer 1m range of large variations in the range of 1m, in the range of 1m, the change is not great, we are in the calculation of 1m depth of the soil water storage below, the soil capacity of the soil weight data can be taken in the soil capacity of the approximate depth of the soil layer of 90-100cm. value.

6.The authors adopted soil desiccation index (SDI) with universal thresholds (wilting point 4.7% and SSM = 60% of field capacity). However, there is no local validation that this parameter is suitable for all micro-topographic soil types in the study site. This has the potential to create over/underestimation in the classification of drought degree.

Response: Thank you very much for your comments and suggestions. In our study, when calculating the soil desiccation index (SDI), the wilting humidity of the soil was used as 4.7%, and the stabilized water holding capacity of the soil was used as 60% of the field holding capacity, and the soil type of Wuji County is a yellow mian soil widely developed on loess parent material after stripping of the zonal black clayey soil, with light loamy texture, a field holding capacity of 20.9%, and a wilting humidity of 4.7%. The soil type in our study area, the Hegou watershed, was found to be light loamy in texture and the soil type was also yellow mian soil through preliminary research. Therefore, we took the value of wilting moisture as 4.7% and field water holding capacity as 20.9% for calculating the soil drying flower index.

7.The conclusions made by the authors tend to Exceed the Evidence. For example, the claim that soil desiccation is “caused by vegetation distribution patterns” has not been supported by statistical analysis of the relationship between vegetation and the SDI index. No regression or correlation is shown. In addition, the description of “natural vegetation recovery” in GU as a cause of soil desiccation is speculative, with no quantitative data on actual root biomass or vegetation structure. How do the authors respond to this?

Response: Thank you very much for your comments and suggestions. Our findings are based on the SDI, and we have put forward some speculations in the discussion section that the reason for the occurrence of soil desiccation is mainly due to the influence of vegetation and other factors, based on the existing studies. According to the results of the existing studies in our group, or according to the results of our survey, Table 1 also shows that the plant biomass is larger in GU, which may be the cause of the moderate desiccation in GU, while the ephemeral gully (EG)�the sink hole (SH), the scarp (SC) and the platform (PL) show mild desiccation, which is likely to be caused by climatic drought.

8.There is no mention of the influence of previous land management, land use history, or potential disturbances (such as human activities, grazing, etc.) that could affect soil moisture or vegetation. I recommend that this is disclosed, albeit briefly.

Response: Thank you very much for your comments and suggestions. The effects of prior land management, land use history, or potential disturbances for the different slope microtopographies studied have been added to the text accordingly, as detailed in L101-L105.

9.The overuse soil moisture value in GU is very high (386.36 mm), but there is no evaluation of whether this value is hydrologically realistic under conditions of 478.3 mm annual rainfall. In addition, the 940 cm thickness of the desiccation layer in PL is quite astonishing, but there is no detailed discussion on how the layer formed to such a depth, even though PL is described as an area with high infiltration and low erosion.

Response: Thank you very much for your comments and suggestions. Our conclusions were based on the SDI values according to the delineation criteria. GU had the largest soil moisture overdepletion of 386.36 mm, which was not only consumed by plant growth, but also due to climate change and other effects such as increased evaporation from soil due to higher temperatures in the Loess Plateau in recent years.

Although the soil dry layer thickness of PL reached 940 cm, the degree of soil desiccation was mild and the desiccation index SDI was the lowest, only 12%. This indicates that soil desiccation occurs in PL but is not severe and is also most likely due to climate change. This requires further research.

10.I did not find a section that explicitly discusses the limitations of the study in the conclusion, even though many large assumptions were used.

Response: Thank you very much for your comments and suggestions. Already added in the discussion, see L317-L318.

---

## [Decision Letter · Decision Letter 1]

2 Sep 2025

Dear Dr. GOU,

Thank you for submitting your revised manuscript to PLOS ONE. After careful consideration, we feel that it has merit but does not fully meet PLOS ONE’s publication criteria as it currently stands. Therefore, we invite you to submit a revised version of the manuscript that addresses the points raised during the review process.

We look forward to receiving your revised manuscript.

Kind regards,

Ziming Yang, PhD

Academic Editor

PLOS ONE

Journal Requirements:

Additional Editor Comments :

Reviewer #1: The authors have made significant improvements to the manuscript. However, I would like to offer some additional suggestions to further strengthen the scientific quality and readability of the manuscript:

Explain that the sample size (n=3 per microtopography) has limitations in capturing the spatial variability of soil moisture in the Loess Plateau. Include a statement that the results should be interpreted with caution, and that further research with a broader sampling design is needed.

If data are available, perform a simple regression or correlation to test the relationship between vegetation parameters (biomass, canopy cover) and SDI. If data are not available, explicitly state that the relationship is still hypothetical based on the literature.

Include brief information (e.g., tables or key data summaries) showing that the characteristics of Hegou represent the Loess Plateau in terms of geomorphology, climate, and soil type.

Review whether the values for overuse soil moisture in GU (386 mm) and dry layer PL (940 cm) are consistent with regional hydrological literature. Include a brief discussion to support the validity of these figures.

Reviewer #3: The manuscript “Soil desiccation of different microtopographies on a slope in the loess area of northern Shaanxi, China” (PONE-D-25-13266R1) addresses an important and relevant topic for soil hydrology and ecological restoration on the Loess Plateau. The revisions made since the first round improve the clarity of the study design and presentation. However, I find that one of my major concerns from the first review remains insufficiently addressed, and this limits the scientific rigor and broader relevance of the paper.

In my initial review, I explicitly requested that the discussion should incorporate and critically engage with the latest studies on climate change impacts on soil moisture and desiccation (including but not limited to doi: [10.2166/wcc.2024.207] and [10.1016/j.catena.2021.105633]). These studies provide essential context for linking the observed soil desiccation patterns to broader regional and global climate-change processes.

In the revised manuscript, the authors briefly mention “climate change” as a contributing factor but do not substantively integrate recent research into their literature review or discussion. This omission weakens the paper, as the findings are not sufficiently situated within the current state of knowledge. Without this, the paper remains largely descriptive and local in scope, rather than demonstrating its broader significance.

Reviewers' comments:

Reviewer's Responses to Questions

**Comments to the Author**

Reviewer #1: All comments have been addressed

Reviewer #3: (No Response)

2. Is the manuscript technically sound, and do the data support the conclusions?

Reviewer #1: Yes

Reviewer #3: Partly

3. Has the statistical analysis been performed appropriately and rigorously?

Reviewer #1: Yes

Reviewer #3: Yes

4. Have the authors made all data underlying the findings in their manuscript fully available?

Reviewer #1: Yes

Reviewer #3: Yes

5. Is the manuscript presented in an intelligible fashion and written in standard English?

Reviewer #1: Yes

Reviewer #3: No

Reviewer #1: Title: Soil desiccation of different microtopographies on a slope in the loess area of northern Shaanxi, China

Manuscript Number PONE-D-25-13266R1

The authors have made significant improvements to the manuscript. However, I would like to offer some additional suggestions to further strengthen the scientific quality and readability of the manuscript:

Explain that the sample size (n=3 per microtopography) has limitations in capturing the spatial variability of soil moisture in the Loess Plateau. Include a statement that the results should be interpreted with caution, and that further research with a broader sampling design is needed.

If data are available, perform a simple regression or correlation to test the relationship between vegetation parameters (biomass, canopy cover) and SDI. If data are not available, explicitly state that the relationship is still hypothetical based on the literature.

Include brief information (e.g., tables or key data summaries) showing that the characteristics of Hegou represent the Loess Plateau in terms of geomorphology, climate, and soil type.

Review whether the values for overuse soil moisture in GU (386 mm) and dry layer PL (940 cm) are consistent with regional hydrological literature. Include a brief discussion to support the validity of these figures.

Reviewer #3: The manuscript “Soil desiccation of different microtopographies on a slope in the loess area of northern Shaanxi, China” (PONE-D-25-13266R1) addresses an important and relevant topic for soil hydrology and ecological restoration on the Loess Plateau. The revisions made since the first round improve the clarity of the study design and presentation. However, I find that one of my major concerns from the first review remains insufficiently addressed, and this limits the scientific rigor and broader relevance of the paper.

In my initial review, I explicitly requested that the discussion should incorporate and critically engage with the latest studies on climate change impacts on soil moisture and desiccation (including but not limited to doi: [10.2166/wcc.2024.207] and [10.1016/j.catena.2021.105633]). These studies provide essential context for linking the observed soil desiccation patterns to broader regional and global climate-change processes.

In the revised manuscript, the authors briefly mention “climate change” as a contributing factor but do not substantively integrate recent research into their literature review or discussion. This omission weakens the paper, as the findings are not sufficiently situated within the current state of knowledge. Without this, the paper remains largely descriptive and local in scope, rather than demonstrating its broader significance.

**Do you want your identity to be public for this peer review?** For information about this choice, including consent withdrawal, please see our Privacy Policy

Reviewer #1: No

Reviewer #3: **Yes: ** Halil Ibrahim Burgan

---

## [Author Response · Author response to Decision Letter 2]

16 Oct 2025

Reviewer #1: The authors have made significant improvements to the manuscript. However, I would like to offer some additional suggestions to further strengthen the scientific quality and readability of the manuscript:

1. Explain that the sample size (n=3 per microtopography) has limitations in capturing the spatial variability of soil moisture in the Loess Plateau. Include a statement that the results should be interpreted with caution, and that further research with a broader sampling design is needed.

Reponse: Thank you very much for your comments and suggestions. Our sampling method is feasible: one end of the 1m iron rod is connected to the handle with a removable screw, and the other end is connected to the soil sampling soil auger with a removable screw, and then 1m iron rod followed by 1m iron rod is connected to each other in such a way that 10 inter-connected ones will be able to pick up soil samples up to a depth of 10 meters. The exact process is shown in the diagram below.

Figure.1 Soil sampling

On one hand, the collection of soil samples presents certain difficulties. On the other hand, the microtopographies under study vary in scale from 1 m² to 100 m². Therefore, only three sampling points were established for each microtopography. During site selection, we prioritized central locations within each microtopography to avoid edge effects near transitional zones between different microtopographies. The conclusions drawn in this study are preliminary, and further data observation is required for verification.

[1]ZHU Q K, ZHANG Y, ZHAO L L, et al. Vegetation Restoration and Near-Natural Afforestation in the Loess Plateau of Northern Shaanxi[M]. Beijing: Science Press, 2012.

2.If data are available, perform a simple regression or correlation to test the relationship between vegetation parameters (biomass, canopy cover) and SDI. If data are not available, explicitly state that the relationship is still hypothetical based on the literature.

Reponse Thank you very much for your comments and suggestions. Currently, the available observational data is insufficient for establishing a simple regression or correlation between vegetation parameters (biomass, canopy cover) and SDI. More extensive experimental observations are required. Furthermore, SDI is influenced not only by vegetation factors but also by topographic and climatic factors. The formation of soil dry layers results from the combined effects of human activities and climate change. In analyzing the relationship between vegetation factors and SDI, we based our discussion on the available data. It is proposed that in microtopographies with higher SDI values, the better soil moisture conditions in shallow layers may have facilitated preliminary vegetation recovery. To ensure better survival, vegetation subsequently utilized deeper soil moisture, ultimately leading to moisture deficit in deep soil layers and an increase in SDI.

3.Include brief information (e.g., tables or key data summaries) showing that the characteristics of Hegou represent the Loess Plateau in terms of geomorphology, climate, and soil type.

Reponse Thank you very much for your comments and suggestions. Hegou is a representative site on the Loess Plateau that our research team has long utilized for eco-hydrological studies. For detailed characteristics of its geomorphology, climate conditions, and soil types, please refer to publications [2-3] from our research team.

[2] Zhao W J, Zhang Y, Zhu Q K, Qin W, Peng S Z, Li P, Zhao Y M, Ma H, Wang Y. Effects of microtopography on spatial point pattern of forest stands on the semi-arid Loess Plateau, China[J]. Journal of Arid Land, 2015a, 7(3):370-380.

[3]Ma H, Zhu Q K, Zhao W J. Soil water response to precipitation in different micro-topographies on the semi-arid Loess Plateau China[J]. Journal of Forestry Research, 2020, 31(1):245-256.

4.Review whether the values for overuse soil moisture in GU (386 mm) and dry layer PL (940 cm) are consistent with regional hydrological literature. Include a brief discussion to support the validity of these figures.

Reponse Thank you very much for your comments and suggestions. Previous studies in our research area (Wuqi County) have documented the occurrence of soil desiccation. Observations from depths of 0–9 m indicate that the soil dry layer has extended beyond 9 meters [4].

[4] Liu G, Wang ZQ, Wang XL. Analysis of dried soil layer of different vegetation types in Wuqi county. Research of Soil and Water Conservation. 2004; 1: 126-129. (In Chinese)

Reviewer #3: The manuscript “Soil desiccation of different microtopographies on a slope in the loess area of northern Shaanxi, China” (PONE-D-25-13266R1) addresses an important and relevant topic for soil hydrology and ecological restoration on the Loess Plateau. The revisions made since the first round improve the clarity of the study design and presentation. However, I find that one of my major concerns from the first review remains insufficiently addressed, and this limits the scientific rigor and broader relevance of the paper.

In my initial review, I explicitly requested that the discussion should incorporate and critically engage with the latest studies on climate change impacts on soil moisture and desiccation (including but not limited to doi: [10.2166/wcc.2024.207] and [10.1016/j.catena.2021.105633]). These studies provide essential context for linking the observed soil desiccation patterns to broader regional and global climate-change processes.

In the revised manuscript, the authors briefly mention “climate change” as a contributing factor but do not substantively integrate recent research into their literature review or discussion. This omission weakens the paper, as the findings are not sufficiently situated within the current state of knowledge. Without this, the paper remains largely descriptive and local in scope, rather than demonstrating its broader significance.

Reponse Thank you very much for your comments and suggestions. The authors have addressed this concern by incorporating discussions on climate change impacts in the designated sections (L316-L323 and L485-L497) of the revised manuscript.

---

## [Decision Letter · Decision Letter 2]

30 Oct 2025

Soil desiccation of different microtopographies on a slope in the loess area of northern Shaanxi, China

PONE-D-25-13266R2

Dear Dr. GOU,

We’re pleased to inform you that your manuscript has been judged scientifically suitable for publication and will be formally accepted for publication once it meets all outstanding technical requirements.

Kind regards,

Ziming Yang, PhD

Academic Editor

PLOS ONE

Additional Editor Comments (optional):

Reviewers' comments:

Reviewer's Responses to Questions

**Comments to the Author**

Reviewer #3: All comments have been addressed

2. Is the manuscript technically sound, and do the data support the conclusions?

Reviewer #3: Yes

3. Has the statistical analysis been performed appropriately and rigorously?

Reviewer #3: Yes

4. Have the authors made all data underlying the findings in their manuscript fully available?

Reviewer #3: Yes

5. Is the manuscript presented in an intelligible fashion and written in standard English?

Reviewer #3: Yes

Reviewer #3: I suggest accepting the manuscript. The authors carefully revised the paper by answering each comment from the last round.

**Do you want your identity to be public for this peer review?** For information about this choice, including consent withdrawal, please see our Privacy Policy

Reviewer #3: **Yes: ** Halil Ibrahim Burgan

---

## [Editor Report · Acceptance letter]

PONE-D-25-13266R2

PLOS ONE

Dear Dr. GOU,

I'm pleased to inform you that your manuscript has been deemed suitable for publication in PLOS ONE. Congratulations! Your manuscript is now being handed over to our production team.

Kind regards,

on behalf of

Dr. Ziming Yang

Academic Editor

PLOS ONE